# Experiences and preferences for psychosocial support: a qualitative study exploring the views of patients with chronic haematological cancers

Rebecca Sheridan ![ORCID],[1] Dorothy McCaughan ![ORCID],[1] Ann Hewison ![ORCID],[1] Eve Roman ![ORCID],[1] Alexandra Smith ![ORCID],[1] Russell Patmore,[2] Debra Howell ![ORCID] [1]

[1]Department of Health Sciences, University of York, York, UK
[2]Queens Centre for Oncology and Haematology, Castle Hill Hospital, Cottingham, UK

**Correspondence to**
Dr Debra Howell;
debra.howell@york.ac.uk

## ABSTRACT

**Objectives** Patients with chronic haematological cancers are often treated on a relapsing-remitting pathway, which may extend for many years. Such diagnoses are associated with uncertainties that often cause anxiety and distress, meaning patients (and families) are susceptible to potentially prolonged emotional difficulties, across the cancer journey. Experiences and preferences regarding psychosocial needs and support over time are relatively unexplored, which this study aimed to address.

**Setting and design** Set within the UK's Haematological Malignancy Research Network (an ongoing population-based cohort that generates evidence to underpin improved clinical practice) a qualitative, exploratory study was conducted, using semistructured interviews. Reflexive thematic analysis was used to assess the interview data via an exploratory, inductive approach, underpinned by the research questions.

**Participants** Thirty-five patients were included with chronic lymphocytic leukaemia, follicular lymphoma, marginal zone lymphoma or myeloma; 10 of whom were interviewed alongside a relative.

**Results** Five themes were identified from the data: (1) accessing support, (2) individual coping behaviour affecting support preferences, (3) divergent and fluctuating thoughts on patient support forums, (4) the role, influence and needs of family and friends and (5) other sources of support and outstanding needs. Findings suggest that patients' individual attitudes towards support varied over time. This also influenced whether support was perceived to be available, and if it was then used.

**Conclusion** This study highlighted the variation in preferences towards psychosocial support among patients with chronic haematological cancers. As patients can live for many years with significant emotional difficulties, they may benefit from frequent monitoring of their psychosocial well-being, as well as signposting to holistic support, if this is needed.

## INTRODUCTION

Haematological malignancies (blood cancers) are a diverse group with many subtypes.[1] Broadly classified as leukaemias, lymphomas and myeloma, these are collectively the fifth most common cancer globally,[2]

## STRENGTHS AND LIMITATIONS OF THIS STUDY

⇒ Comprehensive findings due to inclusion of patients with a range of chronic haematological cancers, demographic characteristics and treatment pathways.
⇒ Robust data collection and analysis conducted by an experienced team of qualitative researchers.
⇒ Transferable findings within the UK and countries with similar healthcare systems.
⇒ Potential recall difficulties due to time since diagnosis, despite the use of diaries/calendars.
⇒ Limited inclusion of relatives/carers (due to patient preferences) and patients under 50 years of age (due to disease rarity); hence the need for further research with these groups.

with increased incidence among ageing populations.[3 4] While some of these malignancies are potentially curable, the majority (around 60%) are not, in which case treatments aim to ameliorate symptoms and slow progression.[5] The four most common chronic haematological cancers (CHCs) are chronic lymphocytic leukaemia (CLL), follicular lymphoma (FL), marginal zone lymphoma (MZL) and myeloma. Some patients with such conditions may be treated from diagnosis; while others will be managed on 'watch and wait', either never requiring treatment, or alternating between treatment and observation with the aim of restoring remission.[6] Treatments may include combinations of chemotherapy, radiotherapy, stem cell transplant and targeted agents.[7–9]

Patients with CHCs may be successfully managed on relapsing-remitting pathways over many years, though symptom burden and loss of functionality can cause increased dependence on others and may have a detrimental effect on quality of life.[10–12] Living with a CHC may be associated with increased emotional and psychological distress, linked primarily to uncertainty concerning disease progression/

relapse, treatment and prognosis, compounded by poor diagnostic understanding, and informational deficiencies.[5] [13–22] Accordingly, the most commonly reported unmet supportive care needs discussed by CHC patients are informational and psychological.[23] The importance of timely psychosocial support during observation, and before, during and after treatment is underscored in a prospective, longitudinal study, which notes an increased risk of patients developing psychological disorders when supportive needs are not met.[24]

CHC patients' experiences of accessing and receiving support remain relatively unexplored, despite the growing significance of this issue due to improved outcomes requiring long-term follow-up; 5-year relative survival estimates range from 48% (myeloma), to 88% (FL).[25] Relevant qualitative literature emphasises the value patients place on social support, which also extends to family and significant others, to enable adjustment and coping.[13] [18] [26] [27] Patients may also rely on peer support groups and healthcare professionals (HCPs), such as clinical nurse specialists (CNSs), for emotional as well as informational support.[18] [27]

Understanding patients' views of seeking and receiving support, and associated barriers and facilitators, is circumscribed by the small-scale of many published studies, predominantly focusing on patients with myeloma.[13] [14] [19] [21] [26] Our own study, therefore, elicited in-depth perspectives from a comparatively large qualitative sample of 35 patients with CLL, FL, myeloma and MZL. Data were explored with the aim of examining the experiences of seeking and receiving emotional and psychological support, from the point of diagnosis onwards, and identifying perceived needs and preferences.

## METHODS

This study is part of a broader programme of work examining the perspectives of CHC patients concerning information needs and treatment decisions.[5] Methods and results are reported according to the Consolidated Criteria for Reporting Qualitative Research guidelines.[28]

### Study design and setting

A qualitative, exploratory study was conducted, using semistructured interviews. It was set within the UK's Haematological Malignancy Research Network (HMRN: https://hmrn.org)[29]; a population-based cohort that was established in 2004 across an area of 4 million people with similar sociodemographic characteristics to the UK as a whole.[30] HMRN is an ongoing collaboration between university academics, National Health Service (NHS) clinicians, and patients and carers, who collectively co-design and conduct research, with the aim of generating evidence that can be used to improve clinical practice, locally, nationally and internationally.

### Sampling

Sampling was purposive; we aimed to capture a range of experiences relating to the topic of interest.[31] Participants with CLL, FL, myeloma or MZL were selected according to the median diagnostic age for each cancer, with variation by gender, time since diagnosis and events on the clinical pathway; an overview of patient characteristics can be found in online supplemental file 1. Refinement of the sampling strategy led to inclusion of patients across broader demographic categories, for example, by including those above and below subtype median diagnostic age.

### Data collection

Checks with patients' clinical teams ensured they were well enough to participate in the study. Identified individuals were sent information and the researcher's contact details to discuss participation. Interviews were conducted by an experienced female researcher (DM), between February and October 2019, at a time and place of the patient's choosing, usually their home, with a friend or relative present if desired. Interviews were continued until data saturation had been reached, defined as the point at which no new insights could be added to the analysis.[32] Participants were invited to ask questions about the study prior to interview, and assurances were given regarding anonymity and confidentiality. Written consent was obtained from all participants, including for the use of direct quotations. Interviews lasted 60–90 min, and were digitally audiorecorded. A semistructured topic guide (online supplemental file 2), developed from the extant research literature,[15–17] [33] [34] and in conjunction with haematology practitioners, was used flexibly to guide discussions, which were subsequently transcribed verbatim, and checked, corrected and pseudonymised by the interviewer.

### Data analysis

Reflexive thematic analysis was used, an approach more akin to a transtheoretical technique, than a methodology predicated on a particular theoretical stance.[35] [36] Analysis was carried out by an experienced team, using an exploratory, inductive approach, underpinned by the research questions. Initial codes were developed by the first author (RS), with further coding and development of themes being a collaborative process involving RS, DM, DH and AH. This was based on active searching for patterns of meaning across the data set, achieved through the dual processes of immersion in the data, and distancing, to allow time for reflection and the development of insights. Regular meetings (RS, DM, DH and AH) focusing on theme review and refinement, supplemented by written memos and reflections, promoted critical thinking, and enhanced reflexivity and interpretive depth. The overall aim was to 'interrogate' the data to arrive at an interpretation that was *'insightful, thoughtful, rich…nuanced'*.[36] Patient quotes are presented in italics and refer to participant

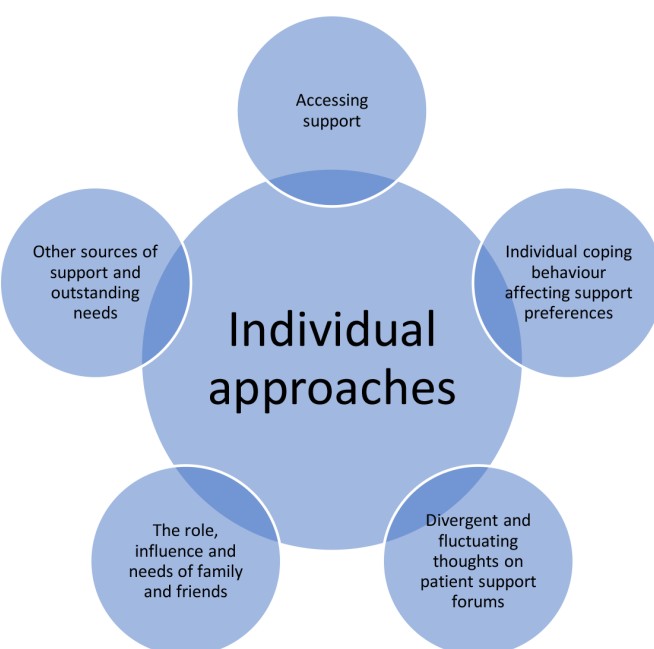

**Figure 1** Diagram of the impact of patients' individual approaches on identified themes.

numbers (eg, P1 for the patient and P1R for the patient's relative).

### Patient and public involvement

Patients and relatives who regularly attended a blood cancer support group were involved in prioritising the study aims, preparing the funding application and disseminating findings. Long-standing links between the research team and members of this group facilitate their active involvement in HMRN's activities, although these patients and relatives did not take part in the interviews.

### RESULTS

Thirty-five patients participated in the study; 10 relatives contributed to interviews. Twelve patients had myeloma, 10 CLL, 8 FL and 5 MZL; treatment pathways varied (online supplemental file 1). Twenty-two patients were aged ≥60 years; and 19 were male.

Five themes were identified: (1) accessing support, (2) individual coping behaviour affecting support preferences, (3) divergent and fluctuating thoughts on patient support forums, (4) the role, influence and needs of family and friends and (5) other sources of support and outstanding needs. Individual preferences were evident in each theme, with the patient's personality impacting their use of support (figure 1).

### Theme 1: accessing support

Support was available from a number of sources, but was impacted by the patient's diagnosis and awareness of the resources available, informed by their own research and/ or HCP knowledge and advice.

#### Subtheme 1.1: varied types of support

Patients reported accessing or awareness of various types of peer support, including face-to-face/online forums, and being introduced to, or knowing, other patients who shared their diagnosis. The majority of face-to-face support groups were not disease specific, but accommodated people with any haematological cancer. Online cancer-specific groups referenced included Myeloma UK, CLL Support Association and also Facebook groups.

Informal support from family and friends was particularly important, while nurses, and, sometimes doctors, also contributed. Patients used NHS and private counselling services and complementary therapies such as acupuncture. Macmillan centres were attended by some, as well as support days and events hosted by charities, such as Myeloma UK.

#### Subtheme 1.2: knowing where to find support

Most patients knew about online and face-to-face support groups, even if they did not want to use these resources. It was not always clear how patients became aware of support groups; some were told by their doctor or nurse (P10, P12, P28), or received information via leaflets (P5). P19 and P3 specifically noted finding websites/forums based in America; P19 found '*it wasn't anything particularly helpful*' due to their non-UK focus.

Some patients contacted others who shared their diagnosis via their general practitioner (GP) or family and friends. For example, P14's GP organised for someone with the same condition to speak to him: '*hearing from somebody who's been through it all and got the badge certainly helped*'. P33 had requested a '*buddy*', something P21R would have welcomed for herself and her relative: '*someone that's like a year in to having this, who would sit down and explain what they discovered and how it works, and what to expect*'.

Some patients (P1, P4) requested counselling, while others (P9, P13) were offered a referral by nursing staff. Nurses also informed some patients (P8, P25) about other measures that might help, such as therapeutic massage. P16 had not been told about Macmillan support centres, but would have found this useful; it was unclear whether other patients were informed about them.

#### Subtheme 1.3: impact of disease subtype on accessing peer support

A number of patients commented on difficulties due to their cancer subtype, such as its rarity (P5, P25, P33). Because of this, P25 felt '*on me [sic] own*' and unsupported, noting that neither her nor her daughter could find any support groups. This could be problematic even for patients who shared the same diagnosis; P5 explained people at her support group all had lymphoma or leukaemia but she didn't perceive herself as having the same disease due to differences in site, clarifying: '*I have [the same cancer] but it's in my stomach*'.

## Theme 2: individual coping behaviour affecting support preferences

The patient's personal approach to their cancer influenced their coping strategies, often resulting in a reluctance to access formal support.

### Subtheme 2.1 perceptions of illness: distancing, distraction, fighting back and maintaining hope

A number of patients didn't want to '*dwell*' on their cancer and discussed measures taken to actively distance or distract themselves. Instead of seeking specific support, such individuals focused on other activities, with P17 saying: '*We have quite a lot of interests, so we're always doing things and I think that's good because it just takes your mind off things*'. P27 and P29 took a philosophical approach to their cancer: '*there's nothing you can do about it…you make the best…of what is available for you*' (P27). Patients did not want to be defined by their condition, or for their condition to '*rule [their] life*' (P3). P5 attended a support group infrequently, as they didn't want their cancer to '*represent*' them. Similarly, P35 commented: '*for some people [at support group] it's like their whole life is having myeloma. I just don't want to be like that*'.

Many patients highlighted specific coping strategies to distract themselves from their CHC, including exercising, dog walking and gardening. Having another focus was important to P3 when trying to accept their diagnosis and deferred treatment (watch and wait): '*[running] made a massive difference because I'm doing something…I'm fighting back*'.

Some people felt they needed to stay positive and maintain hope with respect to their cancer and future outcomes. P24 adopted a '*positive mental attitude*'; P25 described focussing on future events as '*summit [sic] to aim for*'. P13 felt that support group attendance might interfere with their positive approach due to the sharing of negative experiences, explaining: '*I wouldn't want to hear all of the things, you know, "well, it did this to me and it did that to me, and I thought I was on death's door", …I need to keep buoyant and positive about it*'.

### Subtheme 2.2: perceptions of personality

Some patients felt they were not the '*type*' to attend a support group: '*I am just a bit introverted meself [sic]…I'm not a kind of a support group type person…*' (P21). Similarly, P27 described such meetings as '*not for me*', but did acknowledge that this could change in the future. While P10 appreciated that support groups could be useful for others, they found them unnecessary due to them being inherently positive: '*I have a positive attitude and I don't feel I need support*'.

## Theme 3: divergent and fluctuating thoughts on patient support forums

Patients referred to both the significant benefits and also drawbacks of face-to-face and online support forums, with variations noted between individuals.

### Subtheme 3.1: varied perspectives

From a positive perspective, support groups were described as '*very valuable*' (P1); P6 described their group leader as their '*rock*'. Meeting others in a similar situation provided patients with reassurance and comradery: '*you're not on your own but to know somebody else is going through it, and you can tell her your worries*' (P3). P33 also noted that their efforts to maintain positivity within support groups had helped others, saying that members had remarked: '*you've cheered me up and made me feel positive because you're always so positive*'.

While positive experiences were welcomed, hearing negative accounts could be frightening, reflecting patients' desires not to be exposed to these stories (see subtheme 2.1). For example, P35 explained: '*I thought I'd find [support group] terrifying, which I sort of do really because it's like people just talking about their terrible stories a lot of the time*'. P7 highlighted that patients may hear both positive and negative accounts, with the negative making them feel like they were '*in a queue sat there waiting to die*'.

### Subtheme 3.2: changing preferences over time

Patients commented on their decisions regarding when to seek group support, often depending on the status of their cancer. For example, some patients (P4, P8) did not attend face-to-face meetings during treatment due to fatigue, illness or infection risk. Engagement could increase when they needed information or advice, such as when making treatment decisions. As explained by P3, who became more active on internet forums at relapse: '*A lot of people were saying, yeah I was the same as you…I just wanted to forget about it but then when you have something to remind you that you've still got the condition a lot of people start to be more active*'. Some patients on observation didn't feel a support group was necessary as they were '*coping well*' (P17), or because their cancer didn't '*affect [their] lifestyle*' (P20).

### Subtheme 3.3: information provision

Patients generally found support groups and websites a useful source of information, and some noted that they themselves had provided information to others (P6, P13, P19). On websites in particular, patients could choose what information they wanted to engage with, and when. For example, P19 found negative material unwelcome saying: '*it used to scare me*'. P3 highlighted the importance of understanding that '*everyone is different*', explaining: '*you have to decide what suits you, what you feel comfortable with and learn to take the advice that's useful for you and ignore the advice that's less useful*'.

## Theme 4: the role, influence and needs of family and friends

All patients discussed their family and friends, largely referring positively to the support these people provided. The impact of the patients' CHC on family and friends was also considered, the importance of which was recognised by patients.

## Subtheme 4.1: emotional and practical support: reassurance, companionship and the importance of family and friends

Family and friends provided significant emotional support to the majority of patients; for example, P2 commented: *'the people who are close to you are very important because they can keep you going…I think they're very important in reassuring you'*. Patients also acknowledged the importance of practical support, including relatives or friends accompanying them to hospital appointments, reminding them what was discussed, and learning about their condition. P9 explained *'my daughter knows as much about my treatment as I do. She keeps really up-to-date with it and she's with me most treatments'*. Contrastingly, P3 attended appointments alone, because his wife would *'worry herself silly'*, as appointments were *'more stress inducing'* for her than for himself. P34, who attended appointments alone due to his wife's work commitments, acknowledged that, with hindsight, practical support would have been useful.

Emotional and practical input from friends was particularly important for patients without immediate family support. P8, who lived alone, described her friend as *'marvellous'*, explaining: *'I haven't got any family…She was just there for me the whole time'*. P33, whose partner had died, felt the need for practical and emotional support: *'I have to do all my own research, all my own ringing around and sometimes I think I just wish I had that person…I want somebody just to hold my hand and go, I'm going to sort that for you'*. For some, having family did not guarantee support, as exemplified by P25, who relied on a friend to provide emotional and practical help, despite living with relatives.

## Subtheme 4.2: disease characteristics and family dynamics: impact of the cancer, perceived response of others and life context

Patients' decisions about disclosing their CHC impacted available support, and were often influenced by factors relating to their cancer. When patients felt their diagnosis would impact their lives, they often chose to tell family and friends. For example, P22 explained: *'we knew we were always going to always be at hospital…so we took the road that we would tell them'*. In contrast, when patients didn't feel their cancer was noticeably impacting them, for example when on observation, they did not always disclose their diagnosis, thus eliminating opportunities for support. P2 said: *'close family knew about it and close friends and that was it because I could get on with me [sic] life. I could manage it. Nobody needed to know'*. P20 also highlighted the nature of 'watch and wait', explaining they did not inform people about their diagnosis as: *'they wouldn't understand that there's no treatment required'*.

Patients hesitated to disclose their CHC to children for various reasons including their young age, because they were undertaking important exams, and/or because they did not wish to be '*a burden*' (P32). P35 knew she had to tell her children as her diagnosis was *'a big life-changing event'* but was unsure about the timing, explaining: *'I just wanted to know if I could carry on my life without telling them'*. Being on watch and wait allowed P11 to keep their

diagnosis from their children, who they believed were too young to understand/cope.

Even where family and friends were informed about the diagnosis, feelings were not always shared. For example, P7R explained that they did not discuss their worries in order to '*protect*' family. Similarly, P5 felt they had to be strong for others, so were providing, rather than receiving, support. Some patients did not share their feelings about their CHC even with close family, or were reluctant to ask for help, so could not be supported. For example, P2 described shock at their diagnosis, but his wife had said: *'the trouble was you didn't tell me all this so I couldn't be involved'*. P25 described feeling lonely, but wanted to *'make it easier for them [family]'*, admitting: *'I won't ask nobody for help. If somebody, if me [sic] cousin had offered to come, I would have said, "no I'm fine"'*.

Family dynamics could change as a result of the CHC diagnosis. P19 said: *'I didn't feel well and for the first time ever, my family just took full control of everything and normally I'm the one that's like the matriarch'*. However, while the patient was grateful, the diagnosis and changing circumstances were not accepted by everyone: *'[Daughter] wouldn't even talk about it and to this day she doesn't really. She just, you know, she just likes her mum to be mum, and functional'*.

## Subtheme 4.3: the importance of support for family and carers

This was discussed by several patients, with P12 explaining they felt support was '*just as important for the family as it is for the patient*'. Patients appreciated that family are also impacted by their diagnosis; P11 suggested '*it's probably harder on my wife or my children*'. Similarly, P28 remarked that: '*The worry is done by the carers*'. In contrast, P6 acknowledged differences in perceptions of burden between the patient and their family: *'Mentally, [husband] forgets sometimes and he admits—"I forget you've got leukaemia". But it lives with me 24/7. I never forget that I've got it'*.

Patients' families found support in various settings. For example, P12 and P13 described nurses offering support to relatives, with P12 commenting that speaking to the nurse '*kind of settled [wife] a bit*'. P11 encountered another patient, also with myeloma, and explained how their wives then talked and found this useful: '*[Wife (of P11)] was a bit left in the dark she felt, so she really enjoyed talking to [contact's wife] about how she felt*'. Support was also available via dedicated Cancer Centres, with P18's partner accessing counselling and a therapeutic course. P28 praised Myeloma UK for their approach to supporting both patients and carers, including holding sessions specifically for carers at their events.

In contrast, some family members commented that their support needs were unmet. P7R described there being no information about support for relatives, saying '*it's lonely*', and highlighting that the carer also has a *'burden'*. P21R felt unsupported, suggesting counselling should be provided for patients and relatives. Sometimes support was available, but relatives were unwilling to accept it. For example, P19 commented that her daughter really needed support but wouldn't access it: *'the Macmillan*

*nurse was really good and she offered to speak to [daughter], and she tried to ring her and she didn't want to talk about it'.*

## Theme 5: other sources of support and outstanding needs

Support from HCPs and other formal sources, such as counselling services, were discussed by some, with wide variation in the perceived usefulness of these resources. Unmet emotional needs were evident, however, particularly at significant time points in the patient's journey.

### Subtheme 5.1: perceptions of support from HCPs

Nursing staff were considered a valuable source of support, often providing reassurance and encouragement for patients. For example, P14 said: '*[Nurse] has been there for me and guided me, cajoled me, pointed me in the right direction'.* There were differences in the type of support patients perceived nurses as providing; P5 described it as '*medical*' but '*not necessarily psychological*', while P7 said: '*As well as your physical condition [nurses] were bothered about my emotional condition…'.*

A number of patients (P1, P16, P34) reported support from nurses as reliable and always available. In contrast, others were reluctant to access such input; P4 explained: '*I didn't feel I could be ringing [nurse] every day…because obviously she's got lots of other things to do and lots of other people to deal with'.* P4 acknowledged most nurses were '*understanding*', but felt some trivialised her concerns: '*you'd ring up and they'd sort of brush it off'.* P4 and P33 expressed disappointment when nurses reneged on their stated intentions, including referral for counselling, and providing reassurance to family members.

Interactions with nursing staff appeared to vary according to CHC management, with patients on treatment (attending regularly) having greater contact than those being observed on watch and wait (attending less frequently). For example, a number of patients with myeloma (P11, P14, P18, P28), in frequent contact with nurses during treatment, discussed the importance of receiving regular support: '*we've had a lot of dealing with [nurses] and they're spot on*' (P11). Similarly, P3 said: '*The only time I can remember talking to nurses a lot was when I had my treatment and of course they were absolutely superb'.* Preferences were found to vary among those given contact details for nursing staff during periods of observation, with P2 choosing not to use these; while others appreciated this information and felt support was available if needed (P13, P17, P26). Contrastingly, P13 (solely observed since diagnosis) perceived nurses as providing '*holistics…mind, body and soul*' care to those on watch and wait, in order to '*free up the doctors, so they can see the patients that maybe need a little bit of extra time'.*

While most comments about HCP support were about nurses, some patients referred to support from doctors, which was mainly, but not uniformly, perceived positively. P9 felt their doctor was '*very supportive and helpful*' when they were upset about needing chemotherapy, and P12 felt their doctor had '*plenty of empathy*'. In contrast, P4 described their doctors as speaking on a '*consultant level, with no emotions'.* A number of patients highlighted positive interactions with their GPs, with P30 describing theirs as '*fantastic*'. However, others (P6, P13, P23, P25) felt their GP lacked knowledge about their CHC. Three patients mentioned receiving support from other HCPs outwith the NHS, including nurses from Myeloma UK (P29) and Macmillan (P13, P19).

### Subtheme 5.2: counselling and complementary support

Seven patients discussed their experiences of counselling, with perceptions varying markedly. P1 felt this had been useful, facilitating the realisation that they had '*experienced a tremendous trauma*', which had impacted their behaviour in response to treatment and infection fears. Conversely, P25 felt counselling was '*a complete waste of time*' and not what she had expected, explaining: '*To me, a counsellor is somebody that will counsel you and advise you. She never opened her mouth'.* Nevertheless, P25 did feel the service had helped her '*come to terms*' with her diagnosis. P33 felt their psychologist only cared how they felt about their treatment, and not their well-being more generally. P35 described both positive and negative aspects of counselling, explaining that while it was '*helpful*' it could also be upsetting to discuss their feelings.

Four patients (P9, P16, P19, P22) referred to Macmillan services, all positively, though P22 felt these weren't aimed at her as she wasn't '*terminal*'. A number of patients discussed complementary medicines including acupuncture (P9), therapeutic massage (P8, P25) and reflexology (P23). P2 talked at length about the reassurance, support and holistic care provided by the homeopath he had attended for several years, comparing interactions to those with their doctor: '*I used to get different doctors and you just felt as though you were part of a process but…my homeopath I go to, we sit and we talk about my symptoms, how I'm feeling'.*

### Subtheme 5.3: unmet emotional needs

Some patients highlighted unmet emotional support needs. P4, who was waiting for counselling, noted: '*I didn't feel like there was anybody professional shall I say, who I could talk to and just tell them how I felt and that I was really scared…'.* P25 commented that: '*[leaflets] could do [to include] a little bit more on your thoughts. Yeah on your mind, on the emotional side of it',* while P5 felt they would cope with their CHC psychologically, but '*at a price'.*

A number of patients identified key points in the CHC trajectory which caused emotional turmoil, but where they felt their needs were not met. Many highlighted the '*shock*' of diagnosis, which was described as '*a huge emotional rollercoaster*' (P7), while P13 talked about becoming upset when she discovered '*lumps*' which could indicate disease progression; others described feeling shock at being told they would need treatment after a period of observation (P3, P15). Despite the significance of these issues, patients identified a lack of emotional support at these critical junctures in their lives. Concerns could also be continually present, for example patients alluded to the difficulty of living with the uncertainty caused by their cancer

diagnosis: '*it will flare up again…that causes anxiety… you don't know when…it's lurking there*' (P4).

## DISCUSSION

This study provides novel in-depth insights about the experiences of CHC patients in seeking, accessing and engaging with psychosocial support. Many patients relied on support groups, which were generally perceived positively, but there were also significant barriers to their use, including: distress on hearing negative stories, lack of relevant groups due to the rarity of the cancer, and others' poor outcomes interrupting positive approaches to their illness. Family and friends provided psychosocial and practical support, though living with family did not guarantee this. Furthermore, individual patients' personality, outlook on life, and behaviour, impacted on their desire to seek and receive support, and on the family's ability to provide it, with some patients choosing not to disclose their diagnosis. Our study also indicated that positive interactions with HCPs, notably nursing staff, appeared to be largely determined by the patient's pathway; the majority of those who considered nurses as emotionally supportive had received treatment. Finally, both patients and relatives discussed the impact of the CHC on family and friends, stressing the needs of these groups and highlighting gaps in current provision.

The unique nature of CHCs likely impacts on patients' experiences of psychosocial support. For example, patients who did not disclose their diagnosis tended to be those with the most indolent disease, who were being managed on observation, and/or who felt their cancer was not impacting their life and was invisible to others. This lack of disclosure may not have been possible for patients with cancers that are more visible (eg, symptomatic and/or requiring immediate or aggressive treatment). Accordingly, some patients highlighted reluctance to help family and friends understand why treatment was not currently necessary, in contrast to expectations following a cancer diagnosis. In addition, unlike many potentially curable cancers, relapse and further intervention is always possible, and for some diagnoses, is expected. Therefore, the emotional impact of living with a CHC is likely to vary and fluctuate over time, particularly given patients' increasing life expectancy. Thus ongoing, yet potentially intermittent support, which is responsive to the patient's position on their relapsing-remitting pathway, and their associated needs, is especially important.

Our findings are consistent with previous research observing barriers to support group attendance, and thus opportunities for social support. For example, Swash *et al*[18] describe patient difficulties in remaining positive when others with the same diagnosis become increasingly unwell, or die.[18] A reliance on family and friends for social and emotional support has also been reported.[26 27 37] Other research discusses patients withholding cancer related anxiety and distress from friends and relatives to protect them from worry.[13] Generic (eg, Macmillan) cancer support services were rarely mentioned by interviewees in our study and some deemed these inappropriate for CHC. This echoes previous research on CHCs, in which participants didn't always identify themselves as patients with cancer, largely due to their prolonged pathway, lack of obvious symptoms, and associated terminology (eg, haematology not oncology).[18] Limited HCP contact during observation has also been highlighted, with hospital appointments occurring at lengthy (eg, 3–12 months) intervals, thus reducing opportunities for emotional support.[18 23] Resonating with our study, some patients feel uncomfortable raising emotional concerns with HCPs, and feel HCPs do not want to discuss such issues, thereby enhancing barriers to support.[18] The Serious Illness Care Programme (SICP) may be a useful approach to improving communication between patients, families and HCPs.[38] Initially designed as a palliative care intervention, recent research highlights the benefits of introducing SICP earlier in CHC patient journeys to encourage dialogue and enable discussion about fears and concerns associated with the cancer, beyond those related to end of life.[39]

The recent National Cancer Patient Experience Survey (NCPES) found 92.8% of patients with blood cancer said they had a main contact, most often a nurse, to support them during treatment, with most finding these individuals reliable and helpful.[40] HCPs are also in a position to recommend and increase awareness of sources of support,[41] with the NCPES reporting that 89.1% of patients with blood cancers felt hospital staff provided guidance on support groups and other resources, and 78.4% indicating they received the right amount of support with their health and well-being. However, only 29.1% of those who felt they needed emotional support could access this at home post-treatment.[40] This was slightly lower than the overall score for all patients with cancer (31.8%), but is likely to be particularly relevant to those with CHCs, where there is no cure and thus further monitoring, and possibly treatment, is needed. While clinical staff appreciate that CHC patients' emotions may change over time,[42] they may underestimate the impact of these diseases on individual lives and psychological well-being,[15 43] and research is required to fully understand patient needs at various time points. Furthermore, clinical staff, including CNSs, may lack the confidence, time and resources to deal with the psychosocial consequences of cancer,[19 41] or may use their time to focus on physical disease aspects,[44 45] which could result in the under provision of emotional support.[17] Further research should consider the feasibility of HCPs providing psychosocial care in clinical practice, and whether additional resources, including dedicated training, may be required.

Many sources of support discussed by patients are likely to have been impacted by the COVID-19 pandemic, due to lockdown restrictions and infection concerns, given that patients with haematological cancers are particularly vulnerable.[46] This could result in some choosing to avoid face-to-face groups and services, or being encouraged

to do so by HCPs. Online peer support was discussed by patients, and has been found to improve psychosocial well-being in patients with cancer, but is recommended as complementary to face-to-face support rather than a replacement.[47] Providing counselling or similar psychological support via videoconferencing has also been suggested, though barriers have been identified including: concerns regarding confidentiality, technical issues and software requirements (eg, computer and webcam), and whether patients are able to talk openly in private, while in their own home.[48 49]

Previous research has discussed the impact of cancer on patients' families and carers, including worsening psychological well-being, physical health and financial security,[50–53] which may vary throughout illness.[54] Suggestions to improve family outcomes include formally measuring caregiver needs, education and empowerment.[55] A review of measures to assess the impact of caring found that few such tools had been systematically evaluated for cancer populations.[56] There were also gaps, with a limited number covering changing family dynamics (which we found to be important), and many measures being outdated, thus potentially irrelevant, especially for people living longer with cancer.[56] CHCs present specific problems due to their relapsing-remitting nature and the anxiety, repeated upheaval and ongoing uncertainty for patients and families. The loneliness and confusion experienced by some family members in our study suggests that peer support would be useful. A recent feasibility study found one-to-one peer support was acceptable and beneficial for caregivers of those with newly diagnosed haematological cancers,[57] with similar findings for group-based support.[58]

This study has allowed us to capture a breadth of experiences and changing attitudes by interviewing patients at varying intervals post diagnosis, and with different treatment pathways. Although prolonged time since diagnosis may affect recall, referring back to diaries and calendars is likely to have mitigated this; moreover, in many cases, patient accounts were corroborated by relatives. Relatives were not present at every interview, a study limitation, especially with regard to views about support for carers and relatives. Our findings are likely to be transferable within the UK, and in countries with similar approaches to healthcare. However, we note that our sample only includes those over 50 years of age, and while the majority of CHCs are diagnosed in this population, the concerns of younger adults with CHCs may not have been identified. This should be considered in future research, as concerns among this group are likely to differ, possibly including greater emphasis on the potential impact of the CHC on fertility and family life,[59] or the potential for symptoms and/or treatment side effects to impact patients' professional lives, either due to needing to retire or having to adapt their working patterns.[37 60 61]

## CONCLUSIONS

Individual preferences and attitudes resulted in marked differences concerning perceived needs for support, whether and how this was accessed, and the sources utilised. Relationships with others, including HCPs and family members, influenced whether support was considered to be available and if patients drew on it. Preferences were also found to vary within individuals over time, mainly due to the long term, changing nature of CHCs; the need for treatment after observation often leading patients to seek/receive additional support. Patients on observation may have specific difficulties accessing psychosocial support, due to decisions about diagnostic disclosure and limited HCP interactions, which should be considered if such patients are not to be overlooked. Unmet needs were also evident, especially at key points of the clinical pathway, such as diagnosis and relapse, but also more generally, over time. Patients with CHCs may live with distress related to their condition for many years; thus, monitoring emotional needs is of equal importance as physical symptoms, particularly as the latter may never occur. HCPs are ideally placed for this role, as well as appropriate onward sign-posting/referral; although restricted time and resources may necessitate consideration of alternative approaches, and further research may be required to assess feasible possibilities. Our findings underscore the need for patients with CHC to receive support that encompasses their psychosocial needs, and considers their unique treatment pathways and individual lives.

**Acknowledgements** The authors are grateful to the patients and relatives who participated in interviews and shared their experiences with us so openly.

**Contributors** DH, ER, AS and RP designed the study. AS identified potential participants. DM recruited participants and conducted interviews. Transcripts were coded and analysed by RS with input from DM, DH and AH. RS wrote the first draft of the manuscript, which was revised by DH, DM, AH, ER, AS and RP, who also commented on clinical aspects of the study. All authors read and approved the final manuscript. DH is responsible for the overall content as the guarantor.

**Funding** This work was supported by Cancer Research UK (29685), the National Institute for Health Research (RP-PG-0613-2002), and Blood Cancer UK (15037).

**Disclaimer** No funders were involved in the study design, data collection, analysis, interpretation or reporting.

**Competing interests** None declared.

**Patient and public involvement** Patients and/or the public were involved in the design, or conduct, or reporting, or dissemination plans of this research. Refer to the Methods section for further details.

**Patient consent for publication** Not applicable.

**Ethics approval** This study involves human participants and ethical approval was provided by the London, City and East Research Ethics Committee (REC:16/LO/0740). Participants gave informed consent to participate in the study before taking part.

**Provenance and peer review** Not commissioned; externally peer reviewed.

**Data availability statement** No data are available. The datasets analysed during the current study are not publicly available as participants did not provide permission for their entire interview transcripts to be published, and ethical approval was not sought for this. A document containing the qualitative themes, and example codes and quotes is available from the authors on reasonable request.

of the translations (including but not limited to local regulations, clinical guidelines, terminology, drug names and drug dosages), and is not responsible for any error and/or omissions arising from translation and adaptation or otherwise.

**ORCID iDs**
Rebecca Sheridan http://orcid.org/0000-0002-7715-1224
Dorothy McCaughan http://orcid.org/0000-0001-5388-2455
Ann Hewison http://orcid.org/0000-0003-4196-0270
Eve Roman http://orcid.org/0000-0001-7603-3704
Alexandra Smith http://orcid.org/0000-0002-1111-966X
Debra Howell http://orcid.org/0000-0002-7521-7402

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
