## [Reviewer comments · BMJ Open]

ARTICLE DETAILS

TITLE (PROVISIONAL)	Experiences and preferences for psychosocial support: A qualitative study exploring the views of patients with chronic haematological cancers
AUTHORS	Sheridan, Rebecca; McCaughan, Dorothy; Hewison, Ann; Roman, Eve; Smith, Alexandra; Patmore, Russell; Howell, Debra

VERSION 1 – REVIEW

REVIEWER	Tsatsou, Ioanna Hellenic Air Force
REVIEW RETURNED	23-Jan-2023

GENERAL COMMENTS	Dear authors your work is of great importance. The paper is well written and very understandable. I congratulate you on your work. I would suggest some minor changes / additions: -In the introduction I would like to see some more details on supportive care, supportive care needs and social support (since these are themes that emerged from the analysis)-The ethics section is missing -approval from the institution-At the discussion also I would add a few sentences on social support and patient needs, to stress out their importance upon your findings.-Add limitations of the study-Conclusion: a sentence or two on recommendations for future research,
--

REVIEWER	Nielsen, Iben Husted Copenhagen University Hospital
REVIEW RETURNED	31-Jan-2023

GENERAL COMMENTS	Thank you for the opportunity to review this article. It is, overall, a very interesting, well-written article, and investigating the support needs of patients with chronic hematologic cancer is highly relevant. However, I have a few comments to improve the article requiring revisions. In the method section under the patient and public involvement, it is unclear who the patient and relatives are involved in prioritising the research aims, etc. How many were involved, were they recruited from the population cohort, or did they also participate in the interview? Please elaborate on this. The results section presents the findings with six themes and thirteen subthemes, which are too many and may indicate that the themes are premature. I strongly encourage the authors to review the themes and consider refining and merging subthemes. Furthermore, each theme should be introduced and clearly described to determine its essence before presenting subthemes.
--

	In the introduction of the discussion section, the authors provide quite a long summary of findings showing different (and contrasting) patterns, which is nice for the reader. However, preferably this should be integrated into the results, and with a brief summary of findings in the introduction of the discussion. The finding of support needs and preferences for support, specifically in patients on observation, is interesting and provides novel insights. It would be interesting if the authors could discuss these aspects more, especially since healthcare professionals may tend to overlook this patient group. It is nice that the authors discuss different approaches to support, including peer support. I suggest that you also consider discussing the appropriateness of i.e. illness conversations in patients with chronic hematological cancer (see i.e. the article "serious illness conversations in patients with multiple myeloma and their family caregivers—A qualitative interview study" (Myrhoej CB et al 2021)) Furthermore, the important finding of the individualized treatment pathways in CHC patients emphasizes the importance of HCPs being even more attentive to support needs along the trajectory. This is an important finding with clinical relevance and could be clearer in the discussion and conclusion.
--	--

REVIEWER	Boland, Vanessa Trinity College Dublin
REVIEW RETURNED	28-Mar-2023

GENERAL COMMENTS	Thank you for the opportunity to review this qualitative study on an important and emerging area; I agree haematological malignancies are relatively unexplored and deserve further investigation such as this. 1) Introduction The introduction gives an overview of the topic's background and justifies its importance. 2) Data Collection 2.1)"anonymised" - considered using 'pseudonymised' - this is more appropriate for this research as absolute anonymity cannot be achieved. 2.2) "A semi-structured topic guide developed from the research literature". Cite what literature helped guide its development. 3) Data Analysis 3.1) Reflexive Thematic Analysis used. Braun and Clarke approach. 3.2) Author contributions at the end of the paper noted - include this information within data analysis i.e., Independent coding carried out by one author and clarify what input from others means (i.e., verifications?). 3.3) Data saturation is not discussed in the manuscript (mentioned in the supplementary file). 4) PPI Provide examples of how patients/relatives were involved in this research, for example, how they were involved in prioritising research aims or disseminating findings (through a forum... survey etc). 5) Results 5.1) General
---

	Themes 1 to 5 are similarly weighted (word counts, quotations, number of participants etc). Consider theme six as a subtheme within a previous theme as it does not appear as a stand-alone theme after reading the detailed reports for themes 1 to 5. 5.2) Consider reordering the themes; suggest starting with the most 'meaty' or weighted (most prominently discussed by participants); this appears to be theme 5 in its current form. I appreciate this might have been placed lower as it groups together "other sources". The current theme name "Other sources of support and outstanding needs" and its underlying subthemes could be represented as 'supportive care needs' or related umbrella terms. 5.3) Subtheme 5.3 I agree unmet needs are a key area for this group. However, consider explicitly commenting on the unmet needs highlighted by patients to reflect the subtheme name of unmet needs better. The second paragraph outlines emotional turmoil but doesn't mention needs or related terms (concerns, problems etc); consider unmet emotional needs to ease clarity.
--	---

REVIEWER	Cormican, Orlaith University of Galway
REVIEW RETURNED	07-Apr-2023

GENERAL COMMENTS	This was a lovely piece of work to review and well written. I have a few comments which I think will enhance the paper. *On page 6 under sampling, I think it would be worth referring to supplementary files for the reader. *On page 6, under data collection it is mentioned that the interview guide is developed from the research literature, please provide more information. Was this a review done by the team, if so please reference the work. *Regarding the quotations in the text, I would suggest spacing them out from the main text so it is easier to read. It might be worth including the diagnosis beside the participant number too to add context. (I do understand that this is included in the supplementary table also). *Do you think another limitation of the study was the age cohort? The majority of CHC's are in the older population but there are more younger people developing them also and perhaps it is important to mention this. The psychosocial needs of the younger age group are likely to differ because of other factors such as career, raising family, fertility etc. Overall a great piece of work. The focus is definitely changing for this population and the implications for an incurable diagnosis on both the patient and family. I look forward to reading your future work in this area.
--

VERSION 1 – AUTHOR RESPONSE

Reviewer: 1

Dear authors your work is of great importance. The paper is well written and very understandable. I congratulate you on your work. I would suggest some minor changes / additions:

-In the introduction I would like to see some more details on supportive care, supportive care needs and social support (since these are themes that emerged from the analysis)

Thank you for bringing this to our attention. We are constrained by the journal word count but have added further details on these topics.

-The ethics section is missing -approval from the institution

The ethics statement was already in place at the end of the manuscript (pg15), as per journal guidelines.

-At the discussion also I would add a few sentences on social support and patient needs, to stress out their importance upon your findings.

We have now emphasised these aspects in the Discussion.

-Add limitations of the study

Limitations to the study are included in the 'Article summary' and in the last paragraph of the Discussion (pg14). These include potential issues due to: patient recall, relatives not being present at all interviews, omission of patients under 50 years of age.

-Conclusion: a sentence or two on recommendations for future research,

We are grateful for this comment and have added a recommendation for future research in the final section of the Discussion (pg14), where it was better placed.

Reviewer: 2

Thank you for the opportunity to review this article. It is, overall, a very interesting, well-written article, and investigating the support needs of patients with chronic hematologic cancer is highly relevant. However, I have a few comments to improve the article requiring revisions. In the method section under the patient and public involvement, it is unclear who the patient and relatives are involved in prioritising the research aims, etc. How many were involved, were they recruited from the population cohort, or did they also participate in the interview? Please elaborate on this.

We thank the reviewer for this helpful comment. In response, we have added further detail to the section concerning PPIE (pg5).

The results section presents the findings with six themes and thirteen subthemes, which are too many and may indicate that the themes are premature. I strongly encourage the authors to review the themes and consider refining and merging subthemes. Furthermore, each theme should be introduced and clearly described to determine its essence before presenting subthemes.

We have now introduced each theme, as suggested. We have further considered our themes and subthemes and have merged Theme 6 with Theme 4 (pg 10). We do not feel able to further refine or merge our findings as they currently best represent participant data, and were developed iteratively by four of the authors (all of whom were very familiar with the data).

In the introduction of the discussion section, the authors provide quite a long summary of findings showing different (and contrasting) patterns, which is nice for the reader. However, preferably this should be integrated into the results, and with a brief summary of findings in the introduction of the discussion.

We are grateful for this comment and have added descriptive content to the introduction of each theme in the Results.

The finding of support needs and preferences for support, specifically in patients on observation, is interesting and provides novel insights. It would be interesting if the authors could discuss these aspects more, especially since healthcare professionals may tend to overlook this patient group.

Thank you for this positive comment. We have enhanced the text referring to the needs of these patients. Also, a further paper focusing on these patients has recently been accepted, which further addresses these issues.

It is nice that the authors discuss different approaches to support, including peer support. I suggest that you also consider discussing the appropriateness of i.e. illness conversations in patients with chronic hematological cancer (see i.e. the article "serious illness conversations in patients with multiple myeloma and their family caregivers—A qualitative interview study" (Myrhoej CB et al 2021))

Thank you for drawing our attention to this work, which we have now included in our manuscript (pg13).

Furthermore, the important finding of the individualized treatment pathways in CHC patients emphasizes the importance of HCPs being even more attentive to support needs along the trajectory. This is an important finding with clinical relevance and could be clearer in the discussion and conclusion.

We strongly agree with this comment, and have altered some of the text to emphasise the importance of individualised treatment pathways.

Reviewer: 3

Thank you for the opportunity to review this qualitative study on an important and emerging area; I agree haematological malignancies are relatively unexplored and deserve further investigation such as this.

1) Introduction

The introduction gives an overview of the topic's background and justifies its importance.

2) Data Collection

2.1)"anonymised" - considered using 'pseudonymised' - this is more appropriate for this research as absolute anonymity cannot be achieved.

Thank you for drawing our attention to this point; we have amended the text accordingly (pg5).

2.2) "A semi-structured topic guide developed from the research literature". Cite what literature helped guide its development.

Development of the topic guide was based on extant literature pertaining to a range of findings on the information and support needs of patients with chronic haematological cancers, at varying points in their illness trajectories. For example, papers by Atherton et al (2017, 2018) informed development of questions relating to patient preferences for information giving/receiving; and studies by Maher and Vries, 2011; Molassiotis et al, 2011; Evans et al., 2012; Lamers et al., 2013; and Swash et al., 2014 underpinned support needs at various time points. Relevant papers are cited throughout the manuscript, but we accept that it would be useful to reference some of these more explicitly in

connection to development of the topic guide, and have now done this (pg5). Additional references are included:

Atherton K, Young B, Salmon P. Understanding the information needs of people with haematological cancers. A meta-ethnography of quantitative and qualitative research. *Eur J Cancer Care*. 2017;26:e12647. <https://doi.org/10.1111/ecc.12647>

Atherton K, Young B, Kalakonda N, Salmon P. Perspectives of patients with haematological cancer on how clinicians meet their information needs: "Managing" information versus "giving" it. *Psycho- Oncology*. 2018;27:1719–1726. <https://doi.org/10.1002/pon.4714>

3) Data Analysis

3.1) Reflexive Thematic Analysis used. Braun and Clarke approach.

3.2) Author contributions at the end of the paper noted - include this information within data analysis i.e., Independent coding carried out by one author and clarify what input from others means (i.e., verifications?).

As requested, we have added further detail to the data analysis section (pg5).

3.3) Data saturation is not discussed in the manuscript (mentioned in the supplementary file).

Thank you for highlighting this omission. We have now added relevant information, and have cited a related reference (pg5).

4) PPI

Provide examples of how patients/relatives were involved in this research, for example, how they were involved in prioritising research aims or disseminating findings (through a forum... survey etc). We have added further detail to this section (pg5).

5) Results

5.1) General

Themes 1 to 5 are similarly weighted (word counts, quotations, number of participants etc). Consider theme six as a subtheme within a previous theme as it does not appear as a stand-alone theme after reading the detailed reports for themes 1 to 5.

On reflection, we agree with this comment and have consequently edited the manuscript to include Theme six as a subtheme of Theme four, which discusses the role of family and friends (pg10).

5.2) Consider reordering the themes; suggest starting with the most 'meaty' or weighted (most prominently discussed by participants); this appears to be theme 5 in its current form. I appreciate this might have been placed lower as it groups together "other sources". The current theme name "Other sources of support and outstanding needs" and its underlying subthemes could be represented as 'supportive care needs' or related umbrella terms.

We have considered this comment carefully and understand the reviewer's position. However, we feel the current order better represents the significance patients placed on each type of support, whilst the first theme provides useful context for the material that follows.

5.3) Subtheme 5.3

I agree unmet needs are a key area for this group. However, consider explicitly commenting on the unmet needs highlighted by patients to reflect the subtheme name of unmet needs better. The second

paragraph outlines emotional turmoil but doesn't mention needs or related terms (concerns, problems etc); consider unmet emotional needs to ease clarity.

Thank you for drawing our attention to this omission. We have now incorporated these issues into this subtheme more clearly (pg11).

Reviewer: 4

This was a lovely piece of work to review and well written. I have a few comments which I think will enhance the paper.

We would like to thank the reviewer for their positive comments on our manuscript.

*On page 6 under sampling, I think it would be worth referring to supplementary files for the reader.

We have now added this information.

*On page 6, under data collection it is mentioned that the interview guide is developed from the research literature, please provide more information. Was this a review done by the team, if so please reference the work.

We refer the reviewer to our response to a similar comment (see above) from Reviewer 3 (2,2).

*Regarding the quotations in the text, I would suggest spacing them out from the main text so it is easier to read. It might be worth including the diagnosis beside the participant number too to add context. (I do understand that this is included in the supplementary table also).

We appreciate the potential benefits of these suggestions and have considered them carefully. However, we feel that changing the quotations would interrupt readability of the paper, as many small quotes are included within sentences. We also feel that the patient's diagnosis is better alongside other relevant characteristics shown in the table e.g. living circumstances.

*Do you think another limitation of the study was the age cohort? The majority of CHC's are in the older population but there are more younger people developing them also and perhaps it is important to mention this. The psychosocial needs of the younger age group are likely to differ because of other factors such as career, raising family, fertility etc.

We are grateful for this pertinent observation and have edited the text to incorporate age (pg14).

Overall a great piece of work. The focus is definitely changing for this population and the implications for an incurable diagnosis on both the patient and family. I look forward to reading your future work in this area. Thank you!

VERSION 2 – REVIEW

REVIEWER	Tsatsou, Ioanna Hellenic Air Force
REVIEW RETURNED	14-Jun-2023

GENERAL COMMENTS	Page 4: Line 36-38 : by relevant literature you mean qualitative studies? Page 4: Line 55: Further details can be found elsewhere (5):.this sentence does not sound so good. Maybe write it in a different way.
--

	Explain the public involvement? What exactly is the purpose of this paragraph (patient & public involvement)? Thematic analysis is very well written Conclusions: add a few things on implications on clinical practice & future research directions
--	---

REVIEWER	Nielsen, Iben Husted Copenhagen University Hospital
REVIEW RETURNED	21-Jun-2023

GENERAL COMMENTS	Congratulations. You have made notable improvements to the manuscript. Based on these enhancements, I recommend the publication of your paper.
--

REVIEWER	Cormican, Orlaith University of Galway
REVIEW RETURNED	12-Jun-2023

GENERAL COMMENTS	Thank you for making the necessary changes as suggested by the reviewers. It has enhanced the work and provided clarity as well as substance to the paper.
--

VERSION 2 – AUTHOR RESPONSE

Reviewer: 1

Dear authors congratulations on your work.

Page 4: Line 36-38: by relevant literature you mean qualitative studies?
Thank you for highlighting this, we have now clarified it in the text (pg4).

Page 4: Line 55: Further details can be found elsewhere (5): this sentence does not sound so good.
Maybe write it in a different way.
We have removed this sentence but retained the citation to the broader programme of work (pg4).

Explain the public involvement? What exactly is the purpose of this paragraph (patient & public involvement)?
Inclusion of a paragraph detailing Patient and Public Involvement (PPI) is required by the journal (<https://authors.bmj.com/policies/patient-public-partnership/>). The aim of this paragraph is to detail how patients were included in the research process, which is an important aspect of all of our work.

Thematic analysis is very well written
Thank you for this positive comment.

Conclusions: add a few things on implications on clinical practice & future research directions
We previously added a recommendation for future research in the final section of the Discussion (pg14) and have added a further consideration on pg13, as we feel these topics are better placed within the main body of the Discussion, rather than in the Conclusion. Nevertheless, we have added one summary sentence here (p14). Similarly, we consider the implications on clinical practice throughout the Discussion, including referring to the potential use of the Serious Illness Care Program (pg13).